# Cationic Fluorescent Nanogel Thermometers based on Thermoresponsive Poly(*N*-isopropylacrylamide) and Environment-Sensitive Benzofurazan

**DOI:** 10.3390/polym11081305

**Published:** 2019-08-04

**Authors:** Teruyuki Hayashi, Kyoko Kawamoto, Noriko Inada, Seiichi Uchiyama

**Affiliations:** 1College of Nutrition, Koshien University, Hyogo 665-0006, Japan; 2Graduate School of Pharmaceutical Sciences, The University of Tokyo, Tokyo 113-0033, Japan; 3Graduate School of Life and Environmental Sciences, Osaka Prefecture University, Osaka 599-8531, Japan

**Keywords:** fluorescence, imaging, nanogel, temperature, thermometer, thermometry

## Abstract

Cationic nanogels of *N*-isopropylacrylamide (NIPAM), including NIPAM-based cationic fluorescent nanogel thermometers, were synthesized with a cationic radical initiator previously developed in our laboratory. These cationic nanogels were characterized by transmission electron microscopy (TEM), dynamic light scattering (DLS), zeta potential measurements and fluorescence spectroscopy, as summarized in the temperature-dependent fluorescence response based on the structural change in polyNIPAM units in aqueous solutions. Cellular experiments using HeLa (human epithelial carcinoma) cells demonstrated that NIPAM-based cationic fluorescent nanogel thermometers can spontaneously enter the cells under mild conditions (at 25 °C for 20 min) and can show significant fluorescence enhancement without cytotoxicity with increasing culture medium temperature. The combination of the ability to enter cells and non-cytotoxicity is the most important advantage of cationic fluorescent nanogel thermometers compared with other types of fluorescent polymeric thermometers, i.e., anionic nanogel thermometers and cationic/anionic linear polymeric thermometers.

## 1. Introduction

Fluorescent molecular thermometers change their fluorescence properties (e.g., fluorescence intensity, fluorescence quantum yield, fluorescence lifetime and maximum emission wavelength) with temperature [1,2,3,4,5]. Fluorescent polymeric thermometers based on the combination of a thermoresponsive polymer and an environment-sensitive fluorophore [6,7,8,9,10,11,12] are among the most sensitive fluorescent molecular thermometers. Because of their function at the molecular level in aqueous media, these sensitive fluorescent polymeric thermometers have been applied for the thermometry of small subjects, such as microfluids [13,14] and even single living cells [15,16,17,18,19,20,21,22,23,24,25]. Fluorescent polymeric thermometers are classified into two categories of morphology: fluorescent nanogel thermometer with crosslinking units to construct a nano-scaled particle [8,15] and fluorescent linear polymeric thermometer without crosslinking units [6,7,16]. In general, the former is more robust and less interactive with external molecules and ions, whereas the latter is capable of thermometry with a higher spatial resolution, due to its diffusivity.

In 2009, we performed the first intracellular thermometry of live mammalian COS7 (African green monkey kidney) cells using a fluorescent nanogel thermometer that was negatively charged, due to the radical initiator ammonium persulfate (APS) used in its preparation [15]. For intracellular thermometry, the microinjection technique was required for the introduction of anionic fluorescent nanogel thermometers into live COS7 cells. If the ability to spontaneously enter live cells is expected, a fluorescent nanogel thermometer should be positively charged based on the established concept that polycationic structures efficiently support the spontaneous entry of molecules into living cells [26,27,28,29,30]. However, the preparation of positively charged fluorescent nanogel thermometers has not been easily realized because of the lack of cationic radical initiators.

In a recent communication published in 2018 [31], we reported the use of the first cationic radical initiator, 2,2’-azobis-[2-(1,3-dimethyl-4,5-dihydro-1*H*-imidazol-3-ium-2-yl)]propane triflate (ADIP), to prepare cationic nanogels, including a cationic fluorescent nanogel thermometer (Figure 1). In the present article, we describe a detailed experimental procedure to prepare cationic NIPAM nanogels and NIPAM-based cationic fluorescent nanogel thermometers using ADIP and present the comprehensive physical and photophysical properties of these cationic nanogels (i.e., size, zeta potential and/or temperature-dependent fluorescence properties) and the functions of the cationic fluorescent nanogel thermometers in intracellular applications (i.e., ability to enter live cells, distribution in cells, cytotoxicity and response to temperature variation). These experimental protocols and new functional data will complement our earlier communication [31].

## 2. Materials and Methods

Bulletized procedures corresponding to Section 2.2 and Section 2.4 were described in Appendix A.

### 2.1. Materials

NIPAM was purchased from FIJIFILM Wako Pure Chemical Corporation (98%, Osaka, Japan, catalog no. 099-03695) and was purified by recrystallization from *n*-hexane. Cetyltrimethyl­ammonium chloride (CTAC, 95%, catalog no. 087-06032), NaCl (99%, catalog no. 196-01671) and KCl (99.9%, catalog no. 164-13122) were purchased from FIJIFILM Wako Pure Chemical Corporation and were used without further purification. MBAM (99%, catalog no. 148326) and *N*,*N*,*N*',*N*'-tetramethylethylenediamine (TMEDA, 99%, catalog no. T22500) were purchased from Sigma-Aldrich Japan Inc. (Tokyo, Japan) and were used without further purification. D_2_O was obtained from Cambridge Isotope Laboratories (D, 99.9%, Tewksbury, US, catalog no. DLM-4-100). ADIP [31] and DBD-AA [32] were synthesized and purified, as previously reported. Water was purified using a Direct-Q 3 UV system (Merck Millipore, Burlington, MA, USA). Dialysis membranes (MEMBRA-CEL MD34, Molecular Weight Cut Off: 14,000; Pore size: 50 Å) were purchased from Viskase Companies Inc. (Willowbrook, IL, USA).

35-mm Glass dish was purchased from Iwaki/AGC Techno Glass (Shizuoka, Japan, catalog no. 3960-035). Phenol red-free DMEM (Dulbecco’s modified Eagle’s medium, catalog no. 21063-029), FBS (Fetal Bovine Serum, Hyclone, catalog no. SH30910.03), 0.5% Trypsin-EDTA (ethylenediamine­tetraacetic acid, catalog no. 15400054) and MitoTracker Deep Red FM (catalog no. M22426) were purchased from Thermo Fisher Scientific (Waltham, MA, USA). DMEM (catalog no. 08458-45), Na_2_HPO_4_∙12H_2_O (98%, catalog no. 31722-45) and KH_2_PO_4_ (99.5%, catalog no. 28721-55) were purchased from Nacalai Tesque (Kyoto, Japan). HeLa (human epithelial carcinoma) cells were obtained from ATCC (catalog no. CCL-2, Manassas, USA).

### 2.2. Preparation of Cationic Nanogels (NANOGEL-**1** and NANOGEL-**2**) and Cationic Fluorescent Nanogel Thermometers (NANOGEL-**3**~**6**).

NIPAM, DBD-AA, MBAM, TMEDA and/or CTAC were dissolved in 19 mL of water (for the final concentration of each compound in a reaction mixture, see Table 1). Dry argon gas was bubbled through the solution at 70 °C for 30 min to remove the dissolved oxygen. ADIP [31] in water (1 mL) was added to initiate polymerization, and the mixture was stirred using a rod with a paddle at 250 rpm and 70 °C for 1 h under an argon atmosphere. The mixture was then poured into 400 mL of water, and the nanogels were precipitated by a salting-out technique. After purification by dialysis for at least one week, the nanogel dispersions were freeze-dried. The purity of NANOGEL-**1**~**6** was confirmed by ^1^H-NMR measurements with a Bruker AVANCE 400 spectrometer (Billerica, MA, USA) in D_2_O (Appendix A). The yields are indicated in Table 1.

### 2.3. Characterization of NANOGEL-**1**~**6**

TEM images were obtained with a Hitachi H-7100 transmission electron microscope (Hitachi High-Technologies, Tokyo, Japan). A drop of the nanogel solution in ethanol or water (0.01 w/v%, 5 μL) was placed on a formvar-coated copper grid. The specimen was air-dried at room temperature and then examined at an accelerating voltage of 75 kV. The hydrodynamic diameter and zeta potential were measured with a Malvern Instruments Zetasizer Nano ZS (currently Malvern Panalytical, Malvern, UK). The samples were equilibrated for 10 min at each temperature. The amount of fluorescent DBD-AA units in NANOGEL-**3**~**6** was estimated from the comparison of the absorbance of their methanol solution with that of *N*,2-dimethyl-*N*-(2-{methyl[7-(dimethylsulfamoyl)-2,1,3-benzoxadiazol-4-yl]amino}ethyl)propenamide (ε = 11,000 M^−1^ cm^−1^ at 444 nm) as a model compound [33]. The fluorescence spectra of NANOGEL-**3**~**6** were recorded in water and a 150 mM KCl solution at various temperatures using a JASCO FP-6500 spectrofluorometer (Tokyo, Japan) with a Hamamatsu R-7029 optional photomultiplier tube (Hamamatsu, Japan, operating range, 200–850 nm). The sample temperature was controlled using a JASCO ETC-273T temperature controller (Tokyo, Japan).

### 2.4. Introduction of Cationic Fluorescent Nanogel Thermometers into HeLa Cells

The HeLa cells were cultured on a 35-mm glass dish in high-glucose DMEM supplemented with FBS at 37 °C with 5% CO_2_. The cationic fluorescent nanogel thermometers were introduced into the HeLa cells by two different methods: a standard method using cells adhered to a glass-bottom dish and a modified method using suspended cells. For the standard method, DMEM was removed from a glass-bottom dish containing HeLa cells at 30–50% confluency, and the cells were rinsed with 1 mL of 1×PBS (phosphate buffered saline, 10×PBS containing 28.9 g of Na_2_HPO_4_∙12H_2_O, 2.0 g of KH_2_PO_4_, 80.0 g of NaCl and 2.0 g of KCl in a 1 liter solution). Then, PBS was replaced by 1 mL of cationic fluorescent nanogel thermometers in a 5 w/v% glucose solution (0.05 w/v%, 10 μL of a 5 w/v% stock solution in water diluted in 990 μL of a 5 w/v% glucose solution). A 5 w/v% stock solution in water was prepared and incubated at 4 °C at least overnight before the full solvation of nanogels. The dish was incubated at 25 or 37 °C for 5, 10 or 20 min without CO_2_ supply. After incubation, the nanogel solution was removed, and the cells were rinsed with 1 mL of 1×PBS three times. Two milliliters of phenol red-free DMEM was added to the dish before imaging.

For the modified method using suspended cells, HeLa cells cultured in 100 mm culture dishes were rinsed with 1 mL of 1×PBS, then treated with 0.5 mL of a 0.05 w/v% trypsin-EDTA-1×PBS solution and incubated at 37 °C for 3–5 min. The detached cells were suspended in 9 mL of DMEM. One milliliter of the cell suspension was transferred to 1.5 mL tubes and centrifuged at 1200 rpm at 4 °C for 1 min. The collected cells were rinsed twice with 1 mL of 1×PBS, and then suspended in 1 mL of cationic fluorescent nanogel thermometers in a 5 w/v% glucose solution. After incubation at 25 °C for 20 min without CO_2_ supply, the cells were again collected by centrifugation at 1200 rpm at 4 °C for 1 min and rinsed with 1 mL of 1×PBS three times before being suspended in 10 mL of DMEM. Two milliliters mL of cell suspension was added to a 35-mm glass-bottom dish and incubated at 37 °C with 5 % CO_2_ for one or two nights before observation.

### 2.5. Fluorescence Imaging of HeLa Cells

Confocal fluorescence imaging was performed using a TCS-SP5 laser scanning confocal microscope equipped with an HCX PL APO Ibd.BL 63 × 1.4 N.A. oil objective (Leica Microsystems, Wetzlar, Germany). Cells loaded with cationic fluorescent nanogel thermometers were excited by a 458 nm argon laser, then the fluorescence images were acquired through bandpass 500–700 nm in a 1024 × 1024 pixel format, with zoom factors ranging from 1 to 10 and a scanning speed of 400 Hz. The contrast and brightness of the fluorescence images were enhanced using ImageJ with a constant signal intensity ratio. The incorporation efficiencies (%) of NANOGEL-**6** were determined using Equation (1),
Incorporation efficiency (%) = number of cells containing NANOGEL-**6**/number of cells × 100,(1)
in which the total cell number was 183–401, and the cells that showed a fluorescence intensity higher than the threshold (equal to the maximum autofluorescence intensity) were counted as the ″cells containing NANOGEL-**6**″. For the co-visualization of NANOGEL-**6** and mitochondria, the HeLa cells were stained with 50 nM MitoTracker Deep Red FM (MitoTracker DR) in phenol red-free DMEM for 5 min at room temperature and then treated with NANOGEL-**6** by a standard method using adhered cells. A 458 nm argon laser was used to excite NANOGEL-**6**, and a 633 nm HeNe laser was used to excite MitoTracker DR. The fluorescence of NANOGEL-**6** was collected through bandpass 500–600 nm, and the fluorescence of MitoTracker DR was collected through bandpass 645–730 nm.

## 3. Results

### 3.1. Preparation of Cationic Nanogels (NANOGEL-**1** and NANOGEL-**2**) and Cationic Fluorescent Nanogel Thermometers (NANOGEL-**3**~**6**)

The cationic nanogels (NANOGEL-**1** and NANOGEL-**2**) and cationic fluorescent nanogel thermometers (NANOGEL-**3**~**6**) were synthesized by radical polymerization, due to the cationic radical initiator ADIP with the starting materials indicated in Table 1. Crude nanogels were purified by reprecipitation with a salting-out technique and subsequent dialysis for at least one week. The purity of NANOGEL-**1**~**6** was confirmed by ^1^H-NMR measurements (Appendix A). The moderate yields of NANOGEL-**1**~**6** likely resulted from their substantial loss during the reprecipitation (due to high hydrophilicity of NANOGEL-**1**~**6** with cationic surfaces) and the long-term dialysis. The size and surface charge were evaluated by TEM/DLS measurements and zeta potential measurements (Figure 2), respectively, and are summarized in Table 2. The temperature-dependent hydrodynamic diameters of NANOGEL-**1**~**6** determined by DLS clearly show the thermosensitive characteristics of the polyNIPAM units (cf., volume phase transition temperature of the polyNIPAM gel is 32 °C [34]). Sufficient amounts of cationic terminals introduced by ADIP to increase the solubility of nanogels resulted in the isolation of nanogels at a high temperature (45 °C) without aggregation (Table 2). NANOGEL-**1** without fluorescent DBD-AA units was obtained by referring to the concentrations of monomers adopted for the preparation of anionic NIPAM nanogels [15]. Interestingly, the cationic NANOGEL-**2** could be obtained without CTAC, which is a rare example in the preparation of NIPAM nanogels without any surfactant molecules [35]. Then, the cationic fluorescent nanogel thermometers NANOGEL-**3**~**6** were synthesized with various concentrations of an environment-sensitive fluorescent monomer, DBD-AA. The amounts of fluorescent DBD-AA units in NANOGEL-**3**~**6** were converted to the corresponding concentrations when the nanogel concentration was 0.01 w/v% (Table 2).

### 3.2. Fluorescence Responses of Cationic Fluorescent Nanogel Thermometers (NANOGEL-**3**~**6**) in Aqueous Solutions

The fluorescence spectra of NANOGEL-**3**~**6** (0.01 w/v%) in water and 150 mM KCl solution were recorded with changing temperature (Figure 3). The fluorescence intensity ratio at 25 and 45 °C (defined as the fluorescence enhancement (FE) factor) and the maximum emission wavelengths at 25 and 45 °C are listed in Table 2. All samples showed fluorescence enhancements at approximately 32 °C, which is the lower critical solution temperature of PNIPAM nanogels [34]. The heat-induced fluorescence enhancement of NANOGEL-**3**~**6** in 150 mM KCl solution was higher than that in water because of the salting out effects. It is known that salts, including KCl accelerate the dehydration of NIPAM units by hydrogen bonding with their amide groups and by increasing the surface tension of water in the hydration shell around the hydrophobic groups [36,37]. Nevertheless, the salting out effects by KCl on the fluorescence responses of NANOGEL-**3**~**6** were gradually saturated when the concentration of KCl exceeded 50 mM (Appendix A), which is preferred for the use under intracellular conditions (where [K^+^] is approximately 139 mM [38].) As indicated in Figure 3a and Table 2, the maximum emission wavelength at a high temperature (i.e., 45 °C) was remarkably shorter than that at a low temperature (i.e., 25 °C). This temperature-dependent maximum emission wavelength is due to the functional mechanism of NIPAM-based fluorescent polymeric thermometers [6], i.e., the drastic variation of the microenvironment near DBD-AA units with a heat-induced structural change in NIPAM units. As cationic fluorescent nanogel thermometers contain many fluorescent DBD-AA units, the heat-induced fluorescence enhancement became less with a small shift in the maximum emission wavelength between 25 and 45 °C (e.g., NANOGEL-**3** vs NANOGEL-**6** in Table 2). The decrease in sensitivity to the temperature variation is likely due to a structural disturbance of the thermoresponsive nanogels by the relatively bulky DBD-AA units.

### 3.3. Introduction of Cationic Fluorescent Nanogel Thermometers into Mammalian HeLa Cells

The method for introducing cationic fluorescent nanogel thermometers was optimized using the most strongly fluorescent NANOGEL-**6** and HeLa cells as model mammalian cells. One should note that weakly fluorescent nanogel thermometers with fewer DBD-AA units (e.g., NANOGEL-**4**) could not be detected inside the HeLa cells at a low temperature (e.g., 30 °C) with the same laser excitation power and detection sensitivity of the confocal laser scanning microscope used for the detection of NANOGEL-**6**. Figure 4 shows the effect of incubation time at 25 °C on the incorporation efficiency of NANOGEL-**6** when established standard conditions for cationic fluorescent polymeric thermometers (0.05 w/v% in a 5 % glucose solution [28]) were adopted. A high temperature (e.g., 37 °C) accelerated the introduction of cationic fluorescent nanogel thermometers into the HeLa cells, but induced unfavorable aggregation inside the cells. Therefore, we fixed the incubation time and temperature to be 20 min and 25 °C, respectively, in the subsequent experiments. Figure 5 displays the representative transmission and confocal fluorescence images of a HeLa cell containing NANOGEL-**6**, in which the mitochondria were additionally stained by MitoTracker DR. The cationic nanogel thermometer NANOGEL-**6** was detected in the form of dots in the HeLa cells and remained inside them once introduced. While a significant background noise was detected when cationic fluorescent nanogel thermometer NANOGEL-**6** was introduced into adherent HeLa cells, due to the attachment of NANOGEL-**6** on the surface of a glass-bottom dish, treatment of suspended HeLa cells with NANOGEL-**6** could significantly reduce this background noise (Figure 6). This alternative protocol, i.e., introducing NANOGEL-**6** into suspended HeLa cells, is considered for improving the sensitivity of intracellular thermometry by increasing the signal-to-noise ratio.

### 3.4. Functions of Cationic Fluorescent Nanogel Thermometers inside HeLa Cells

The functions of NANOGEL-**6** in HeLa cells were examined as a representative cationic fluorescent nanogel thermometer. Similar to the response in aqueous solutions, the cationic fluorescent nanogel thermometer NANOGEL-**6** introduced into HeLa cells showed remarkable fluorescence enhancement when the temperature of the culture medium was increased (Figure 7).

Another unique property of cationic fluorescent nanogel thermometers prepared by ADIP is non-cytotoxicity. As demonstrated in Figure 8, the HeLa cells containing NANOGEL-**6** were capable of dividing in a similar manner to those without staining (i.e., control). The fluorescence response of NANOGEL-**6** to temperature variation was confirmed for the HeLa cells incubated for one day after the introduction of NANOGEL-**6** (Figure 8).

## 4. Discussion

In the present study, we demonstrated the synthesis of cationic nanogels (NANOGEL-**1**~**6**) and presented a functional assessment of cationic fluorescent nanogel thermometers (NANOGEL-**3**~**6**). One of the important advantages of fluorescent polymeric thermometers is the capability of functional integration by introducing additional units into a macromolecule. In the case of NANOGEL-**3**~**6**, the NIPAM units were assumed to be sensitive to temperature variations, whereas the DBD-AA units produced a fluorescence signal. The crosslinker MBAM units provided the robustness of nanoparticles. The positive charges on the surface, which originated from the radical initiator ADIP, enabled spontaneous entry into mammalian cells. Non-cytotoxicity was also a consequence of this cationic surface of the nanogel. Targeting to organelles by attaching a specific signal on the surface or signal normalization by introducing a second reference fluorophore [39] will be expected to improve in the future.

Table 3 summarizes both the advantages and disadvantages of the four types of fluorescent polymeric thermometers ever developed. Now, we can fill the last empty column in Table 3. In general, cationic fluorescent thermometers are more useful than anionic ones because the former can be introduced into mammalian cells under mild conditions without a microinjection technique so that a large number of samples can be treated. Fluorescent nanogel thermometers and fluorescent linear polymeric thermometers are complementary: fluorescent nanogel thermometers show low toxicity to live cells, but the spatial resolution in intracellular thermometry is low, while fluorescent linear thermometers show opposite characteristics. Recently, biological and even medical researchers have begun to utilize fluorescent polymeric thermometers in their own studies [40,41,42]. The cationic fluorescent nanogel thermometers developed in this study will contribute to the field of intracellular thermometry, due to their remarkable non-cytotoxicity, which will enable long-term observations.

## Figures and Tables

**Figure 1 polymers-11-01305-f001:**
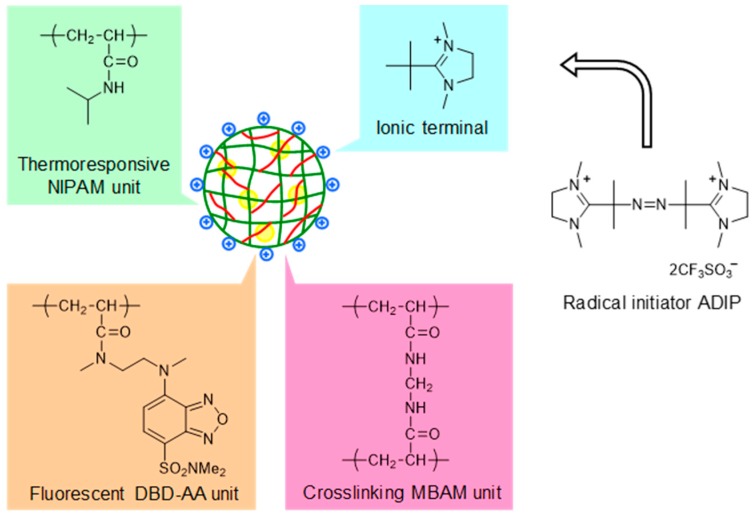
Cationic fluorescent nanogel thermometer prepared with cationic radical initiator ADIP (2,2’-azobis-[2-(1,3-dimethyl-4,5-dihydro-1*H*-imidazol-3-ium-2-yl)]propane triflate). NIPAM: *N*-isopropylacrylamide, DBD-AA: *N*-{2-(7-*N*,*N*-dimethylaminosulfonyl-2,1,3-benzoxadiazol-4-yl)methylamino}ethyl-*N*-methylacrylamide, MBAM: *N*,*N*’-methylenebisacrylamide.

**Figure 2 polymers-11-01305-f002:**
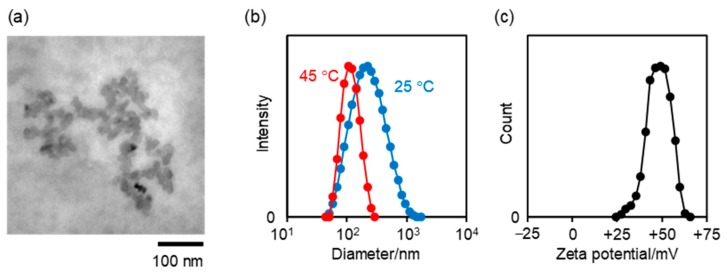
Characterization of NANOGEL-**1** as a representative. (**a**) TEM image; (**b**) size distribution measured by DLS (0.001 w/v% in water at 25 and 45 °C); (**c**) zeta potential distribution (0.1 w/v% in water at 45 °C).

**Figure 3 polymers-11-01305-f003:**
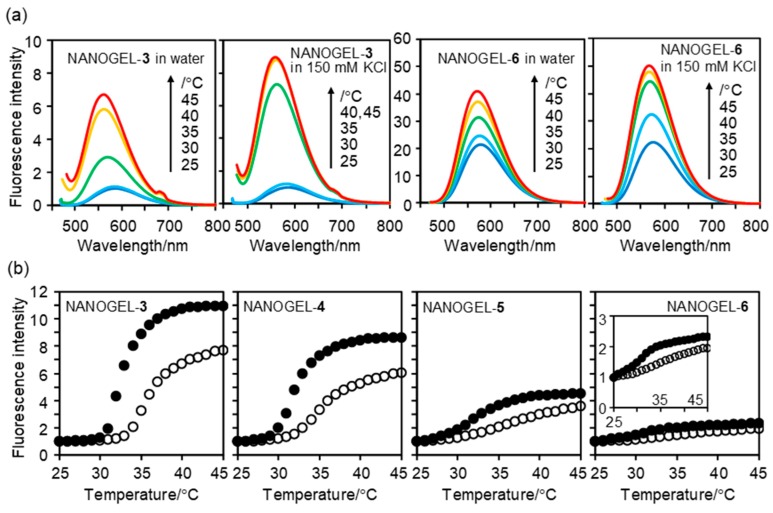
Fluorescence response of NANOGEL-**3**~**6** to temperature variation. (**a**) Temperature-dependent fluorescence spectra of NANOGEL-**3** and NANOGEL-**6** in water and 150 mM KCl aqueous solution. The vertical units in (**a**) are identical.; (**b**) Fluorescence intensity of NANOGEL-**3**~**6** in water (open circle) and 150 mM KCl aqueous solution (closed circle) at λ_em_ at 45 °C (see Table 2). The inset for NANOGEL-**6** is vertically expanded. Concentration of NANOGEL-**3**~**6**: 0.01 w/v%; excitation: 456 nm. The shoulders at approximately 684 nm in the fluorescence spectra at 40 and 45 °C in panel (**a**) were due to the scattered excitation light.

**Figure 4 polymers-11-01305-f004:**
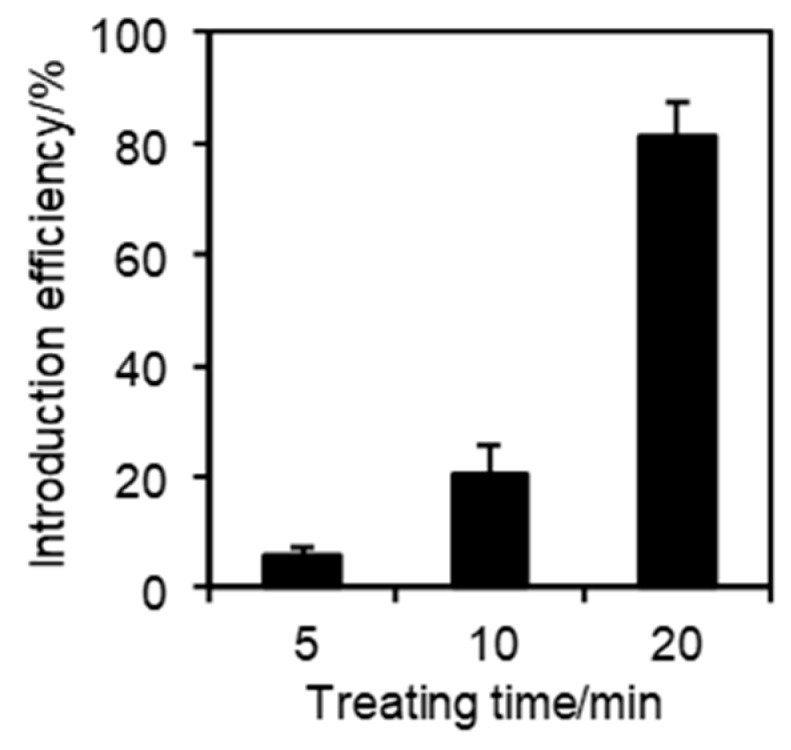
Effects of the incubation time on the incorporation efficiency. The HeLa cells were treated with NANOGEL-**6** (0.05 w/v%) in 5 % glucose solution at 25 °C for the indicated durations. The vertical bars represent the s.d. evaluated by at least four independent experiments.

**Figure 5 polymers-11-01305-f005:**
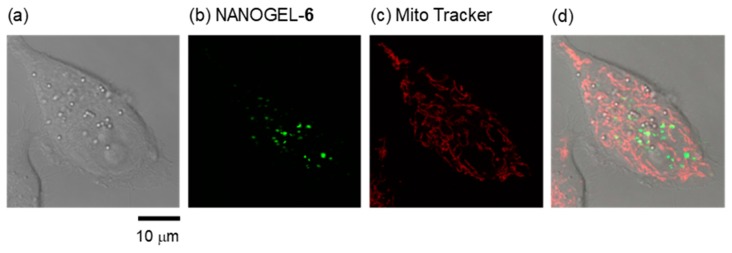
Cationic fluorescent nanogel thermometer NANOGEL-**6** in a live HeLa cell. (**a**) Transmitted light image; (**b**) confocal fluorescence image of NANOGEL-**6** (excitation: 458 nm; emission: 500–600 nm); (**c**) confocal fluorescence image of mitochondria visualized with Mito Tracker DR (excitation: 633 nm; emission: 645–730 nm); (**d**) merged image.

**Figure 6 polymers-11-01305-f006:**
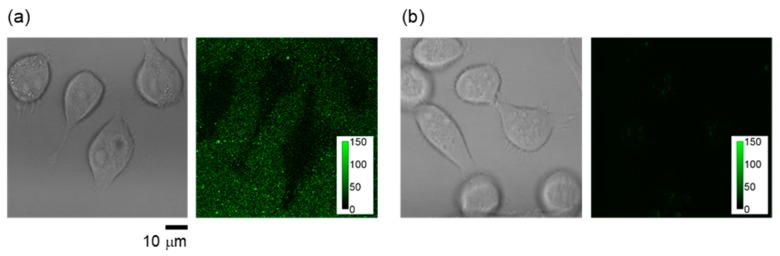
Reduction of background noise in the fluorescence image of HeLa cells with NANOGEL-**6** by the modified incorporation method using suspended cells. (**a**) Transmitted light image (left) and confocal fluorescence image (right, excitation: 458 nm; emission: 500–600 nm) at 37 °C when NANOGEL-**6** was introduced into adherent HeLa cells attached on a dish.; (**b**) Transmitted light image (left) and confocal fluorescence image (right, excitation: 458 nm; emission: 500–600 nm) at 37 °C when NANOGEL-**6** was introduced into suspended HeLa cells before scattering on the dish. The confocal fluorescence images were focused on the glass surface of the dish. Color bars in the fluorescence images indicate the fluorescence intensity (arbitrary unit).

**Figure 7 polymers-11-01305-f007:**
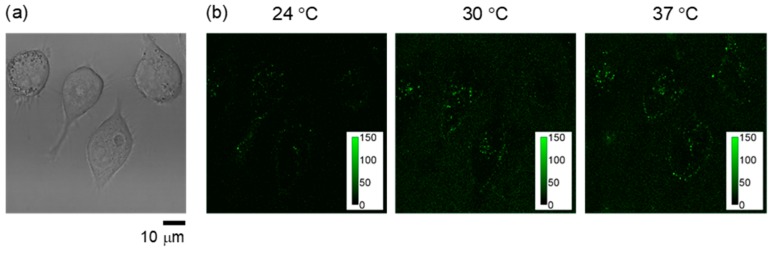
Fluorescence response of NANOGEL-**6** in live HeLa cells. (**a**) Transmitted light image at 37 °C; (**b**) confocal fluorescence image (excitation: 458 nm; emission: 500–600 nm) at 24 °C (left), 30 °C (middle) and 37 °C (right). Color bars in the fluorescence images indicate the fluorescence intensity (arbitrary unit).

**Figure 8 polymers-11-01305-f008:**
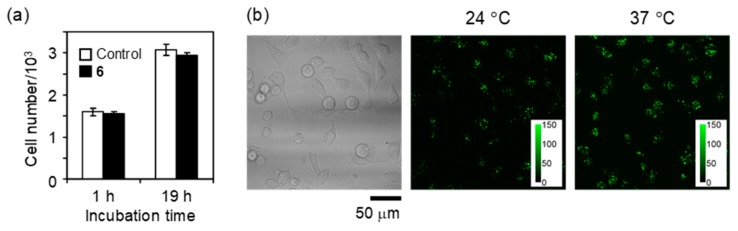
Non-cytotoxicity and long-term functionality of NANOGEL-**6** in live HeLa cells. (**a**) Growth of HeLa cells containing NANOGEL-**6** and none (control) (mean ± s.d.); (**b**) Transmitted light image at 37 °C (left) and confocal fluorescence image (excitation: 458 nm; emission: 500–600 nm) at 24 °C (middle) and 37 °C (right) at 24 h after the introduction of NANOGEL-**6** into the cells. Color bars in the fluorescence images indicate the fluorescence intensity (arbitrary unit).

**Table 1 polymers-11-01305-t001:** Preparation of cationic nanogels (NANOGEL-**1** and NANOGEL-**2**) and cationic fluorescent nanogel thermometers (NANOGEL-**3**~**6**).

Name	Final Concentrations in the Reaction Mixture						Yield (%)
	NIPAM (mM)	MBAM (mM)	DBD-AA (mM)	TMEDA (mM)	ADIP (mM)	CTAC (mM)	
NANOGEL-**1**	100	1		2.9	28	1.9	39
NANOGEL-**2**	100	1		2.9	28		14
NANOGEL-**3**	100	1	0.1	2.9	28	1.9	35
NANOGEL-**4**	100	1	0.2	2.9	28	1.9	47
NANOGEL-**5**	100	1	0.5	2.9	28	1.9	28
NANOGEL-**6**	100	1	1	2.9	28	1.9	46

**Table 2 polymers-11-01305-t002:** Physical and photophysical properties of cationic nanogels (NANOGEL-**1** and NANOGEL-**2**) and cationic fluorescent nanogel thermometers (NANOGEL-**3**~**6**).

Name	Diameter (nm) ^1^		Zeta Potential (mV) ^2^	[DBD-AA unit] (μM) ^3^	FE ^4^	λ_em_ (nm)		FE ^4^
	at 25 °C	at 45 °C			in Water	at 25 °C	at 45 °C	in 150 mM KCl sol.
NANOGEL-**1**	270 ± 31	123 ± 1.4	+47.8 ± 0.9	―	―	―	―	―
NANOGEL-**2**	284 ± 7.8	117 ± 2.7	+56.9 ± 0.7	―	―	―	―	―
NANOGEL-**3**	232 ± 25	90.2 ± 7.2	+37.3 ± 0.3	1.09	7.72	589	561	10.90
NANOGEL-**4**	311 ± 24	151 ± 11	+45.0 ± 1.0	2.42	6.03	585	563	8.60
NANOGEL-**5**	230 ± 32	133 ± 4.0	+48.8 ± 0.2	6.64	3.58	582	567	4.50
NANOGEL-**6**	314 ± 20	157 ± 27	+53.5 ± 0.2	12.0	1.95	577	570	2.30

^1^ Determined by DLS. The av ± s.d. of peak values in triplicate measurements. ^2^ At 45 °C. The av ± s.d. of peak values of triplicate measurements. ^3^ When the nanogel concentration is 0.01 w/v%. ^4^ Fluorescence enhancement factor calculated as fluorescence intensity at λ_em_ at 45 °C divided by that at 25 °C.

**Table 3 polymers-11-01305-t003:** Comparison of fluorescent polymeric thermometers for intracellular thermometry.

	Morphology	Nanogel	Linear Polymer
Charge ^1^			
Cationic		- moderate spatial resolution (single-cell level) ^2^ - spontaneously entering into cell - non-cytotoxicity	- high spatial resolution (200 nm) ^3^ - spontaneously entering into cell - relatively high cytotoxicity ^4^
Anionic		- moderate spatial resolution (single-cell level) ^2^ - requiring microinjection - relatively low cytotoxicity	- high spatial resolution (200 nm) ^3^ - requiring microinjection - relatively high cytotoxicity ^4^

^1^ Neutral fluorescent polymeric thermometers cannot be utilized for intracellular thermometry, due to serious aggregation under the high ionic strength inside living cells. ^2^ The average temperature of a single cell can be monitored. ^3^ The temperature distribution inside a cell can be imaged. ^4^ Interfering with cell division.

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
