# Peer review of "Cationic Fluorescent Nanogel Thermometers based on Thermoresponsive Poly(N-isopropylacrylamide) and Environment-Sensitive Benzofurazan"

_polymers, 2019, doi:10.3390/polym11081305_

Round 1

Reviewer 1 Report

The present paper reports the synthesis of cationic fluorescent nanogel thermometers and their application in live imaging of cells. The presented work is an extension of a previous communication by the same authors (Angew.Chem.Int.Ed. 2018, 57, 5413 –541), providing a more detailed synthetic protocol and characterization of the obtained nanogels. However given the focus on the technical part of the synthesis and characterization, some points need to be addressed before publication:

2. Materials and Methods

Please add numbering on the paragraphs of this section

Since you claim in the introduction that this article is focused on the experimental protocol and the characterization of the nanogels, you should add more details in material and methods. A bullet points list for the protocol steps could be more helpful to the reader.

Are the reagents commercially available or synthesised in your lab? What is the purity? Can you comment on the very low yield of some of the prepared nanogels?

3. Results

Please add the NMR spectra of the prepared nanogels in Supplementary Information.

3.2. Fluorescence responses of cationic fluorescent nanogel thermometers (3–6) in aqueous solutions

In Fig 3.b the fluorescent response in water in comparison to the one in 150mM for 6 should be added, as done in Fig.3a

3.4. Functions of cationic fluorescent nanogel thermometers inside HeLa cells

In Fig.8b the temperature indicate on top of the middle panel (30 C) does not correspond to the one indicated in the caption (24 C). Please correct the figure

Reviewer 2 Report

Authors demonstrated the synthesis of cationic nanogels functional assessment for the use as a nanogel thermometers. The methods and results are presented with clarity and substantiated with intuitive experimental designs and data. However, the evidence presented needs better presentation, especially when it comes to the main results shown: fluorescence images in figures 6,7,8.  Few suggestions to improve the MS are listed below. Apart from these minor comments, I find the study exciting and promising to the fluorescence imaging community.

Suggestions:

1) The fluorescence images presented are entirely invisible in the current color scheme. Use http://fiji.sc/ to change the colormap and use color bars to denote the intensity scale. 

2) Use a more coherent naming convention than a numeric. "NANOGEL-6" instead of "6".

3) Line 196: the "salting out" effect in nanogels needs to be elaborated more. This environmental sensitivity is a critical factor, especially when it comes to cytoplasmic distribution. 
